# Rollover Stability of Heavy-Duty AGVs in Turns Considering Variation in Friction Coefficient

**Weijie Fu** [1,2,3,*] , **Xinyu Wang** [2,3] **and Xinming Zhang** [1,2,3]

1   School of Mechatronic Engineering and Automation, Foshan University, Foshan 528001, China; zxm@cust.edu.cn
2   School of Mechatronic Engineering, Changchun University of Science and Technology, Changchun 130022, China
3   Ministry of Education Key Laboratory for Cross-Scale Micro and Nano Manufacturing, Changchun 130022, China
*   Correspondence: weijie.fu@hotmail.com

**Abstract:** This study analysed the impact of turning rate, the centre of mass height, and road adhesion coefficient on the rollover stability of heavy-duty automated guided vehicles (AGVs) using a multi-body dynamics simulation model. The lateral deflection angle of the centre of mass was used as a metric to evaluate the rollover behaviour of the AGVs, and the results were obtained quantitatively. The findings showed that the turning rate had the largest impact on AGV rollover stability, followed by the centre of mass height, while the road adhesion coefficient had the least impact. Despite having the lowest impact, the road adhesion coefficient was one of the key factors contributing to AGV slippage, and severe slippage could easily lead to rollover incidents. To further evaluate the rollover behaviour of the AGVs, a lateral-load transfer rate (LTR) index was derived from the change in wheel load generated by the lateral tilt angle during steering. The range of LTR values was determined for different ranges of turning rate, the centre of mass height, and road adhesion coefficient. The results indicated that the range of LTR values for turning rate was 0.23–0.45, for the centre of mass height was 0.323–0.393, and the minimum value of 0.337 was obtained for a road adhesion coefficient of 0.6.

**Keywords:** heavy-duty AGVs; sway stability; lateral-load transfer rate; orthogonal test analysis

## 1. Introduction

In the field of automated material handling, Automated Guided Vehicles (AGVs) have been a popular tool since their introduction in the 1950s. Heavy-duty AGVs, designed for transporting heavy cargo, have larger sizes and higher centres of mass than traditional vehicles, resulting in a higher likelihood of rollover incidents during steering. In order to improve the safety of heavy-duty AGVs during steering, E. Dahlberg proposed a theoretical method for calculating the rollover threshold value of AGVs by analysing the mathematical model of AGV rollover [1]. This highlights the rollover model's crucial role in the AGV stability study.

Thomas and Woodrooffe studied the load transfer between the AGV wheels prior to rollover [2]. They introduced the "Lateral-Load Transfer Ratio (LTR)" index to determine AGV rollover hazard. The LTR concept is based on the conclusion that manoeuvring stability varies with changes in the centre of mass, wheel pressure, and wheel width. By analysing the forces and moments acting on the AGV, the vertical load on the wheel during horizontal curve travel can be modelled [3].

Sampson and Cebon investigated the roll stability of heavy AGVs and the use of active roll control to improve performance [4]. They showed that the optimal control goal is to achieve equal normalized load transfer on all critical axes and to roll the vehicle inward to the maximum angle allowed by the suspension. Optimal controllers using the linear quadratic Gaussian (LQG) method and LTR were developed for turning flexible

unit vehicles and heavy AGVs travelling at a constant speed, resulting in significant improvements in rolling stability (30–40%).

Aleksander Hac investigated the influence of AGV suspension parameters and tire deformation on vehicle roll [5]. A. Bruce Dunwoody applied an active suspension system to AGVs, increasing the rollover threshold and lowering the centre of gravity [6]. D.J.M. Sampson and D. Cebon developed an active suspension rollover control system for AGVs, which can reduce the steady-state and transient load transfer rate and improve the rollover threshold compared to passive suspension systems [7]. Christopher J. Constantine reduced the vehicle roll angle by changing the suspension stiffness without adding extra energy during the process [8,9].

Torbic et al. studied the dynamics of AGVs with higher degrees of freedom, considering different geometric features of the AGV model, such as horizontal and vertical curves [10,11]. Kordaniet evaluated the stability of AGVs when manoeuvring through various geometric features and analysed the adequacy of wheel friction in multiple cases, redesigning each influencing factor for stability [12–15].

Despite extensive research on the factors contributing to AGV rollover, there is a need to examine the evolving trend of these factors during the steering process of heavy-duty AGVs and quantify the extent of change. In this study, we investigate the impact of changes in turning rate, the centre of mass position, and road adhesion coefficient on the rollover of AGVs during steering. Utilizing the theoretical method of "AGV dynamic rollover queue", a simulation model of a heavy-duty AGV is established. By analysing the main factors that affect AGV rollover, we produce a trend graph showing the influence of each factor on rollover. The impact of heavy-duty AGV rollover factors is also evaluated. This research not only helps prevent heavy-duty AGVs from overturning but also ensures their safe operation during steering, enhances their carrying efficiency, and holds significant implications for AGV steering and control.

## 2. The Mechanism of Steering Rollover of AGV

### 2.1. Model of AGV Rollover

The critical factors influencing the rollover stability of a heavy-duty Automated Guided Vehicle (AGV) are the driving speed and road conditions. Figure 1 illustrates the rollover model that takes into account the displacement of the AGV's centre of mass during the steering process. The AGV under examination is equipped with a 5-ton capacity stainless steel stand that stands 5 m tall. The chassis features four steerable wheels that are driven by independent servo motors. The equation for moment balance at the point of contact (P) between the outer wheel and the ground of the AGV's sideswipe model during the steering bend is depicted in Figure 1 [16].

The factors that determine the stability of an AGV during overload tipping include the travelling speed and road conditions. Figure 1 shows a rollover model that considers the deviation of the suspension centre of mass during the turning process. The AGV is applied to a large support for 5 tons of 201 stainless steel with a height of 5 m, and the chassis has four helm wheels independently driven by servo motors, as shown in Figure 1. By taking the force torque balance at the contact point p between the outer wheel of the AGV rollover model and the ground during the turning of a bend, the following equation can be obtained [16]:

$$m_s a_y h_g - m_s g \left[ B/2 - \Phi \left( h_g - h_r \right) \right] + F_{zi} B = 0 \tag{1}$$

$$\Phi = R_\Phi \frac{a_y}{g} \tag{2}$$

When $F_{zi} = 0$, the AGV starts to tip over and can no longer maintain balance within the rolling plane. The threshold of tipping over when the AGV starts to tip over is given by the following Equation (3):

$$R_\Phi = \frac{B_g - 2a_y h_g}{2a_y h_g (1 - h_r/h_g)} \tag{3}$$

where $R_\Phi$ is the roll rate of the AGV, $m_s$ is the total mass of the AGV; $B$ is the wheelbase of the AGV; $a_y$ is the lateral acceleration of the AGV; $h_g$ is the height of the centre of mass from the ground; $h_r$ is the height of the roll centre from the ground; $F_{zi}$ is the vertical reaction force received by the inner wheel of the AGV; and $\Phi$ is the roll angle of the centre of mass of the AGV.

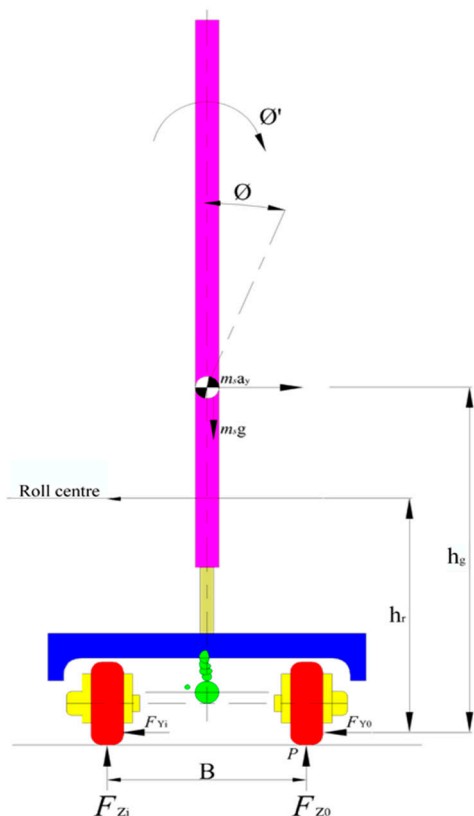

**Figure 1.** The Steering Rollover Model of a Heavy-duty AGV.

*2.2. Dynamic Rollover Model for Heavy-Duty AGV*

According to the AGV steering-rollover mechanism described above, a dynamic rollover model of AGV turning motion is established, as shown in Figure 2. In the analysis process, it is necessary to make some condition assumptions to reduce the influence of irrelevant variables on the rollover model during the turning process [16]: the effect of the steering system is not considered; the mass of AGV is concentrated at the centre of gravity; AGV moves uniformly at a constant speed along the *x*-axis; the swinging and rolling motion of AGV are taken into account; air resistance is ignored; the changes of steer wheel duty caused by ground camber force and helm wheel characteristic changes are neglected; the rolling centre is close to the ground. The established three-degree-of-freedom model of AGV is shown in Figure 2, with each helm wheel driven individually by a servo motor.

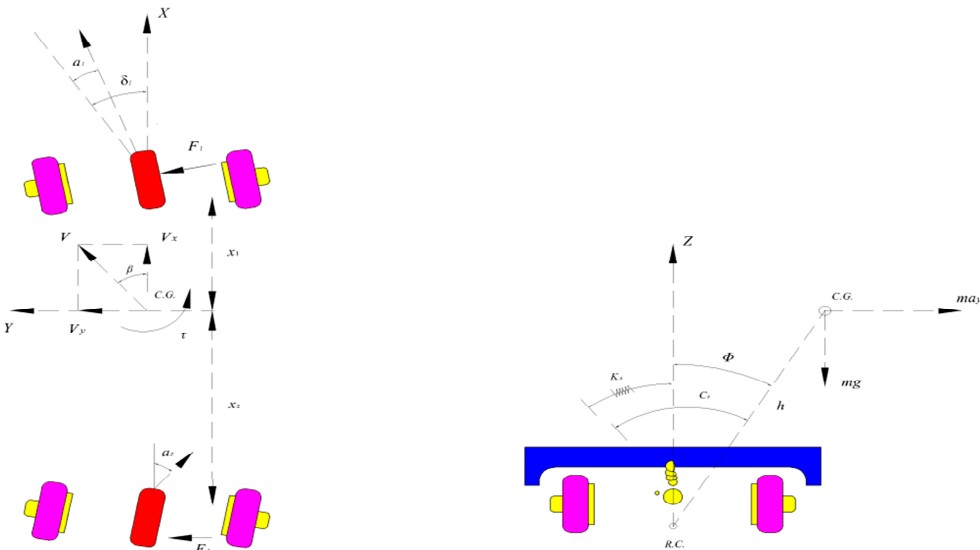

**Figure 2.** The dynamic rollover model of AGV.

The lateral force $F_1$ is generated between the helm wheel and the ground due to the increase of the wheel side deflection, and the lateral force $F_1$ generates a transverse moment to the centre of mass of the AGV, which causes the AGV to swing sideways. The lateral force $F_2$ is generated between the helm wheel and the ground to balance the tendency of the AGV to swing sideways, and the lateral acceleration $a_y$ causes the AGV to roll sideways. Through the analysis of the above heavy-duty AGV motion state, the following differential equations of motion can be established.

The lateral motion equation is:

$$\sum F_y = ma_y = F_1 + F_2 + mh\ddot{\phi} \tag{4}$$

The yaw motion equation is:

$$\sum M_z = I_z\ddot{\psi} = x_1 F_1 + x_2 F_2 \tag{5}$$

The roll motion equation is:

$$\sum M_z = I_{xeq}\ddot{\phi} = ma_y h - c_z\dot{\phi} + (mgh - k_s)\phi \tag{6}$$

where $I_z$ is the moment of inertia of AGV around the $Z$ axis, $I_{xeq}$ is the moment of inertia of AGV around the roll centre, and $a_y$ is the lateral acceleration; $V_x$ and $V_y$ represent the moving speed of AGV along the $X$-axis and $Y$-axis respectively. $r$ represents the yaw rate of AGV; $\phi$ represents the sideslip angle of AGV centroid; $\psi$ represents the yaw angle of AGV; $\phi$ represents the AGV roll angle; $\delta$ represents the wheel angle; $m$ represents AGV quality; $h$ represents the distance from the centre of mass to the roll centre; $k_s$ and $C_s$ represent the suspension roll stiffness and roll damping, respectively.

Equations (4)–(6) can represent the AGV in the steering process due to the lateral force to produce moments, the lateral force of the AGV is unbalanced, the AGV produces lateral acceleration $a_y$. and there is a lateral acceleration $a_y$ of the AGV, will produce a certain lateral roll angle $\phi$, in the lateral, transverse, lateral roll three directions of motion at the same time, with the increase of lateral force and lateral acceleration, the moment increases, at this time the AGV makes the lateral rollover motion.

According to the AGV dynamic rollover model, it can be seen that the above motion differential equation conforms to the parallel-axis theorem:

$$I_{xeq} = I_x + mh^2 \tag{7}$$

where $I_x$ is the moment of inertia of AGV around the *X*-axis.

The lateral acceleration ay of AGV along the *Y*-axis is:

$$a_y = \dot{V}_y + rV_x \tag{8}$$

Considering the steering angle δ is generally small, wheel lateral force $F_1$, $F_2$ can be calculated by the following equation:

$$F_1 = k_1\alpha_1 + k_2\alpha_2 \tag{9}$$

where $k_1$ and $k_2$ are the equivalent lateral deflection stiffness of the front and rear wheels, respectively, and are taken as positive values. When the wheel lateral deflection angle is small, the lateral deflection angles $a_1$ and $a_2$ satisfy the following relations.

$$\alpha_1 = \delta - \frac{V_y + rx_1}{V_x}, \alpha_2 = -\frac{V_y + rx_2}{V_x} \tag{10}$$

The equation for the lateral deflection angle of the centre of mass of a heavy-duty AGV is $\beta = V_y/V_x$ In summary we have:

$$\left\{ \begin{array}{l} mV_x\left(\dot{\beta} + r\right) = \frac{(k_1+k_2)I_{xeq}}{I_x}\beta - \frac{(x_1k_1+x_2k_2)I_{xeq}}{I_xV_x}r + \frac{mh(mgh-k_s)}{I_x}\phi - \frac{mhc_s}{I_x}\dot{\phi} + \frac{k_1I_{xeq}}{I_x}\delta \\ I_z\dot{r} = -(x_1k_1 + x_2k_2)\beta - \frac{x_1^2k_1+x_2^2k_2}{V_x}r + x_1k_1\delta \\ I_x\ddot{\phi} = -(k_1 + k_2)h\beta - \frac{(x_1k_1+x_2k_2)}{V_x}r + (mgh - k_s)\phi - c_s\dot{\phi} + hk_1\delta \end{array} \right\} \tag{11}$$

Equation (11) can describe the dynamic three degrees of freedom motion differential equation of the heavy-duty AGV in the three directions of lateral, yaw, and roll during the steering process. The lateral force of the inner and outer wheels changes instantaneously during the steering process, and the motion state in all directions is also constantly changing. The roll stiffness $k_s$ and roll damping $C_s$ of the AGV suspension can have a certain resistance effect on the roll motion of the AGV, but when the lateral acceleration increases to a certain limit value, the AGV roll angle will increase to exceed the limit value and cause a rollover. This dynamic model provides a theoretical basis for the mathematical model of omnidirectional mobile AGV.

### 2.3. The Omnidirectional Kinematics Model of Heavy-Duty AGV

2.3.1. Mathematical Model of a Heavy-Duty AGV with Four Helm Wheels

The motion model of heavy-duty AGV describes the relationship between the rate of change of the state quantities (speed, attitude, etc.) of the AGV and the control quantities (speed and steering of the wheels). Based on the motion model of the heavy-duty AGV, the world coordinate system $\sum XOY$, the AGV coordinate system $\sum X_rOY_r$, the rudder wheel coordinate system $\sum X_{wr}OY_{wr}$, and the remaining parameters are shown in the Table 1.

**Table 1.** Heavy-duty AGV parameters description table.

| Symbols | Implication | Symbols | Implication |
| --- | --- | --- | --- |
| $V_1$ | Right front helm wheel speed | $V_2$ | Left front helm wheel speed |
| $V_3$ | Left rear helm wheel speed | $V_4$ | Right rear helm wheel speed |
| $\beta_1$ | Angle between right front helm wheel and AGV coordinate system | $\beta_2$ | Angle between left front helm wheel and AGV coordinate system |

**Table 1.** *Cont.*

| Symbols | Implication | Symbols | Implication |
|---------|-------------|---------|-------------|
| $\beta_3$ | Angle between left rear helm wheel and AGV coordinate system | $\beta_4$ | Angle between right rear helm wheel and AGV coordinate system |
| $W_1$ | Right front helm wheel angular velocity | $W_2$ | Left front helm wheel angular velocity |
| $W_3$ | Left rear helm wheel angular velocity | $W_4$ | Right rear helm wheel angular velocity |
| $r_1$ | Right front helm wheel rotation radius | $r_2$ | Left front helm wheel rotation radius |
| $r_3$ | Left rear helm wheel rotation radius | $r_4$ | Right rear helm wheel rotation radius |
| $V$ | Heavy-duty AGV speed | $R$ | Heavy-duty AGV rotation radius |
| $\omega$ | Heavy-duty AGV angular velocity | $I$ | Heavy-duty AGV rotary centre |
| $X_i$ | The lateral distance of the helm wheel from the centre position | $Y_i$ | The longitudinal distance between the helm wheel and the centre position |
| $\alpha$ | Heavy load AGV rotary centre abscissa | $b$ | Heavy load AGV rotary centre ordinate |

### 2.3.2. Mathematical Model for Positive Kinematics Analysis of Heavy-Duty AGV

The heavy-duty AGV has four helm wheels that can be turned and driven independently. This structure not only makes the heavy-duty AGV bear a large duty but also theoretically realizes the curved motion of any radius in any direction of the moving AGV so that it has high spatial adaptability. In the actual operation process, it is necessary to model and analyse the curved motion [17].

In the modelling process, only its motion-related parameters are considered for the sake of intuitive kinematics, and the following assumptions are made for heavy-duty AGV motion.

(1)    The heavy-duty AGV is a rigid body.
(2)    The running ground of the heavy-duty AGV is horizontal and of suitable smoothness.
(3)    The motion speed of heavy-duty AGV is low and there is no air resistance.
(4)    The driving wheel of heavy-duty AGV has good contact with the ground, and the driving wheel does pure rolling.

The motion of heavy-duty AGVs can be defined as the path motion problem of AGVs [18]. According to the four helm wheels kinematic analysis, the overall equation for a heavy-duty AGV helm at a point can be expressed as:

$$\xi(t) = [x(t)y(t)\theta(t)]^T \tag{12}$$

Figure 3 shows the kinematic model of heavy-duty AGV curvilinear motion, with the geometric constraint that the helm heavy-duty AGV makes rotary motion around the instantaneous centre of rotation at every moment. That is the velocity vectors of the helm heavy-duty AGVs at any given moment all intersect at the instantaneous centre of rotation. As can be seen from the figure, heavy-duty AGV is located in the world coordinate system, helm heavy-duty AGV four helm wheels can be regarded as a parallel mechanism AGV coordinate system can be along the wheel *i* and virtual wheel after rotation and translation movement back to its own original location, respectively. The motion of the AGV coordinate system along wheel *i* is as follows: translate $|X_i|$ along the $X_r$ axis, then

translate along the $Y_r$ axis to move to the centre of wheel *i*, rotate $\beta_i$ angular value at the centre of wheel *i*, and the AGV coordinate system coincides with the helm wheel coordinate system, then translate $Fr_{Ii}$ along the helm wheel coordinate system respectively, translate the AGV coordinate system to the centre of rotation I, reverse $\beta_i$ angular value around the centre of rotation, and in $X_r$ and $Y_r$ axes translations to move the helm wheel coordinate system to the origin *Or*, then the coordinate transformation equation of the motion of the AGV coordinate system along wheel *i* is expressed as:

$$
{}^{Or}T_{Or} = \begin{bmatrix} 1 & 0 & 0 & X_i \\ 0 & 1 & 0 & Y_i \\ 0 & 0 & 1 & 0 \\ 0 & 0 & 0 & 1 \end{bmatrix} \begin{bmatrix} \cos\beta_i & -\sin\beta_i & 0 & 0 \\ \sin\beta_i & \cos\beta_i & 0 & 0 \\ 0 & 0 & 1 & 0 \\ 0 & 0 & 0 & 1 \end{bmatrix} \begin{bmatrix} 1 & 0 & 0 & r_{Ii} \\ 0 & 1 & 0 & 0 \\ 0 & 0 & 1 & 0 \\ 0 & 0 & 0 & 1 \end{bmatrix} \begin{bmatrix} \cos\beta_i & \sin\beta_i & 0 & 0 \\ -\sin\beta_i & \cos\beta_i & 0 & 0 \\ 0 & 0 & 1 & 0 \\ 0 & 0 & 0 & 1 \end{bmatrix} \begin{bmatrix} 1 & 0 & 0 & -a \\ 0 & 1 & 0 & -b \\ 0 & 0 & 1 & 0 \\ 0 & 0 & 0 & 1 \end{bmatrix} \tag{13}
$$

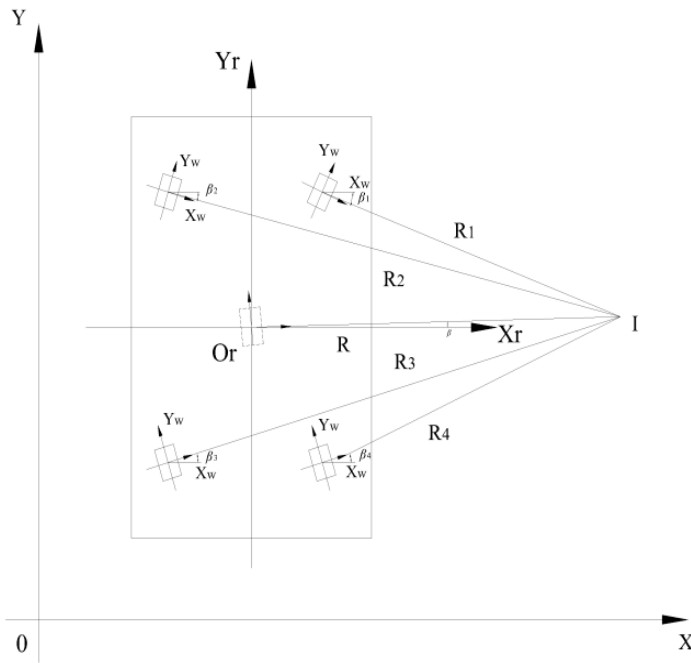

**Figure 3.** Heavy-duty AGV curve steering motion diagram.

The motion of the AGV coordinate system $\sum X_r O Y_r$ along the virtual wheel is as follows: the AGV coordinate system rotates $\beta$ angle value at its own origin to make it coincide with the virtual wheel coordinates, then translates $r_I$ along the *X*-axis coordinates of the virtual wheel to the centre of rotation, reverses $\beta$ angle, and then translates along the $X_r$ and $Y_r$ axes respectively to return to the initial position, then the coordinate transformation equation of the motion of AGV coordinate system along the virtual wheel is expressed as:

$$
{}^{Or}T_{Or} = \begin{bmatrix} \cos\beta & -\sin\beta & 0 & 0 \\ \sin\beta & \cos\beta & 0 & 0 \\ 0 & 0 & 1 & 0 \\ 0 & 0 & 0 & 1 \end{bmatrix} \begin{bmatrix} 1 & 0 & 0 & r_i \\ 0 & 1 & 0 & 0 \\ 0 & 0 & 1 & 0 \\ 0 & 0 & 0 & 1 \end{bmatrix} \begin{bmatrix} \cos\beta & \sin\beta & 0 & 0 \\ -\sin\beta & \cos\beta & 0 & 0 \\ 0 & 0 & 1 & 0 \\ 0 & 0 & 0 & 1 \end{bmatrix} \begin{bmatrix} 1 & 0 & 0 & -a \\ 0 & 1 & 0 & -b \\ 0 & 0 & 1 & 0 \\ 0 & 0 & 0 & 1 \end{bmatrix} \tag{14}
$$

The calculation results are shown below.

$$
{}^{Or}T_{Or} = \begin{bmatrix} 1 & 0 & 0 & -a + r_{Ii}\cos\beta_i + X_i \\ 0 & 1 & 0 & -b + r_{Ii}\sin\beta_i + Y_i \\ 0 & 0 & 1 & 0 \\ 0 & 0 & 0 & 1 \end{bmatrix} \tag{15}
$$

Variation of the above equation yields.

$$^{Or}T_{Or} = \begin{bmatrix} 1 & 0 & 0 & -a + r_I \cos \beta \\ 0 & 1 & 0 & -b + r_I \sin \beta \\ 0 & 0 & 1 & 0 \\ 0 & 0 & 0 & 1 \end{bmatrix} \tag{16}$$

From the above equation, according to the principle of equality of its own rotation matrix, the substitution of the relevant parameters gives.

$$\begin{aligned} a &= \frac{\sin \beta_1 \cos \beta_2 X_1 - \cos \beta_1 \sin \beta_2 X_2 - (Y_1 - Y_2) \cos \beta_1 \cos \beta_2}{\sin(\beta_1 - \beta_2)} \\ b &= \frac{\sin \beta_1 \sin \beta_2 (X_1 - X_2) - \cos \beta_1 \sin \beta_2 Y_1 + \sin \beta_1 \cos \beta_2 Y_2}{\sin(\beta_1 - \beta_2)} \end{aligned} \tag{17}$$

The following equation is obtained according to the above figure.

$$\beta = \arctan\left(\frac{a}{b}\right) \tag{18}$$

$$r_I = \sqrt{a^2 + b^2} \tag{19}$$

$$r_{Ii} = \cos \beta_i (a - X_{ri}) + \sin \beta_i (b - Y_{ri}) \tag{20}$$

Then the angular velocity equation of heavy-duty AGV is shown below.

$$\omega_i = \omega = \frac{V_i}{r_{Ii}} \tag{21}$$

The centre instantaneous speed of the heavy-duty AGV is shown below.

$$V = r_I \times \omega \tag{22}$$

The angle between the AGV coordinate system and the world coordinate system can be specified as $\theta$. The velocity direction of the heavy-duty AGV in the world coordinate system is shown below.

$$\beta_0 = \beta + \theta \tag{23}$$

The velocity component of the heavy-duty AGV in the AGV coordinate system can be expressed by the following equation.

$$\begin{cases} V_x = (V_i/r_i)r_i \cos \beta \\ V_y = (V_i/r_i)r_i \sin \beta \\ \omega = V_i/r_i \end{cases} \tag{24}$$

Transferring the above equation to the whole world coordinate system, the positive kinematic model of the heavy-duty AGV is represented by Equation (24).

$$\begin{cases} \dot{x} = V_x \cos \beta_0 - V_y \sin \beta_0 \\ \dot{y} = V_x \sin \beta_0 + V_y \cos \beta_0 \\ \dot{\theta} = \omega \end{cases} \tag{25}$$

### 2.3.3. Inverse Kinematics Analysis of the Model

Heavy-duty AGV inverse kinematics analysis is to solve the velocity and angle values of the helm in the case of specified AGV travel speed V and radius R. In the process of heavy-duty AGV movement, correct inverse kinematics modelling is the basis of AGV helm movement and control operation, taking the four helm wheels mode as an example, the centre of rotation is fixed as the X-axis outside the coordinate system AGV, as shown in the Figure 4.

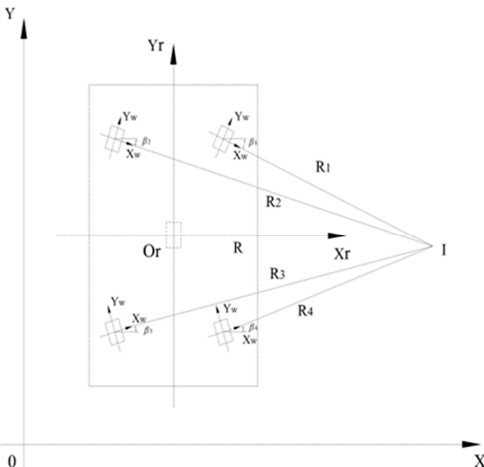

**Figure 4.** Heavy-duty AGV four-wheel steering mode.

The central velocity of the four helm wheels heavy-duty AGV is V, its turning radius is R, and the velocities of the four helm wheels are $V_1$, $V_2$, $V_3$, and $V_4$, and their angles are $\beta_1$, $\beta_2$, $\beta_3$, and $\beta_4$, respectively. then it is described in Figure 4.

$$\tan \beta_1 = \frac{-Y_1}{R - X_1} \tag{26}$$

Variation of Equation (26) yields Equation (27).

$$\beta_1 = \arctan\left(\frac{-Y_1}{R - X_1}\right) \tag{27}$$

Applying the same method, Equations (28) and (29) are obtained.

$$\beta_2 = \arctan\left(\frac{-Y_1}{R + X_1}\right) \tag{28}$$

$$\beta_3 = \arctan\left(\frac{Y_1}{R + X_1}\right) \tag{29}$$

The operating speed of a four-helm wheel heavy-duty AGV can be expressed by Equations (30) and (31).

$$V_1 = V_4 = R_1 \times \omega = \sqrt{(R - X_1)^2 + Y_1^2} \tag{30}$$

$$V_2 = V_3 = R_2 \times \omega = \sqrt{(R + X_1)^2 + Y_1^2} \tag{31}$$

According to the above analysis, the speed and angle of each helm can be derived when the heavy-duty AGV is rotating around any point outside the AGV as the centre of rotation, which proves the omnidirectional mobility of the heavy-duty AGV steering around any point and lays the foundation for the steering motion simulation route design and steering sway analysis of the heavy-duty AGV.

## 3. Heavy-Loaded AGV Rollover Simulation Analysis

### 3.1. Simplification of AGV Model

The heavy-duty AGV studied is applied to an automated machine with a duty capacity of 5 tons and a significant structural size, which can realize the omnidirectional movement of the carried cargo centre of mass on the heavy-duty AGV. This model mainly includes front and rear suspension, front and rear wheels, AGV chassis, load plate, power system, steering system and braking system. Due to the complexity of the AGV structure, expressing it not

only increases the complexity of the model but also increases the simulation time, reducing the simulation efficiency. Therefore, without affecting the simulation analysis of the AGV bend rollover does not affect the AGV structure to simplify:

(1) Simplify the AGV into a mass point, so that the AGV mass is concentrated in one point;
(2) Ignoring the influence of suspension characteristics when considering the AGV model as a rigid object to perform force analysis;
(3) Ignoring the asymmetry of the left and right tires and the front and rear axles;
(4) Ignoring the effect of longitudinal motion of the AGV on rollover during steady-state steering and driving;
(5) The influence of the dynamic characteristics of the AGV in the pitch direction on rollover is not considered.

### 3.2. Simulation Modelling

Using ADAMS simulation modelling, a simplified model is used as the simulation model to simulate and analyse its rollover process to study the AGV rollover risk. In the modelling process, the template characteristics of each subsystem (mass, rotational inertia, positioning parameters.) need to be corrected, and the constraint relationships between the components need to be established. The simplified model of heavy-duty AGV is shown in Figure 5 and the simulation parameters are shown in Table 2.

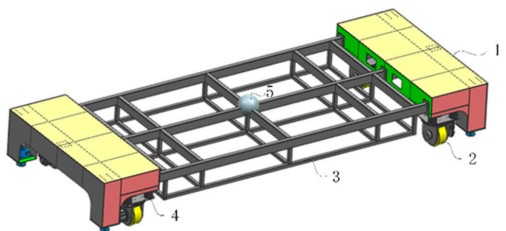

**Figure 5.** Simplified model of heavy-duty AGV. (1) AGV chassis, (2) helm mechanism, (3) AGV carrier plate, (4) Braking system, and (5) Mass centre ball.

**Table 2.** Simulation parameters of AGV steering motion.

| Variable Names | Symbols | Numerical Values | Units |
|---|---|---|---|
| AGV quality | $M$ | 5 | t |
| Duty | $M_S$ | 10 | t |
| Total height | $h$ | 5 | m |
| Rotational inertia around the *x*-axis | $I_X$ | 21,300 | kg×m$^2$ |
| Distance from the centre of mass to the front axle | $a$ | 3.5 | m |
| Distance from the centre of mass to the rear axis | $b$ | 3.5 | m |
| Sway stiffness | $K_\phi$ | 90,672 | N/rad |
| Sway damping | $C_\phi$ | 5677 | N/rad |
| Rotational inertia around the *z*-axis | $I_Z$ | 58,893 | kg×m$^2$ |
| Turning radius | $r$ | 6 | m |
| Wheelbase | $l$ | 3.590 | m |
| Sum of lateral deflection stiffness of two front wheels | $k_1$ | −60 | kN/rad |
| Sum of lateral deflection stiffness of both rear wheels | $k_2$ | −60 | KN/rad |
| The inertia of rotation around the centre of the lateral roll | $I_{xeq}$ | 64,952 | kg×m$^2$ |

### 3.3. Simulation Process of Heavy-Duty AGV

The following figure shows the simulation process of the motion of the simplified model of heavy-duty AGV at different moments. For example, the AGV's turning speed was set from 0 to 100, the linear speed is 10.47 m/s, and with this rate in the radius of 6 m

on the curve for circular motion, from the figure can be seen, the simplified model in the design of the track for circular motion, simplified. The model has an apparent rollover phenomenon at 8 s, and the rollover is more evident at 9.92 s, and the simulation is finished. The simplified simulation model of heavy-duty AGV is shown in Figure 6.

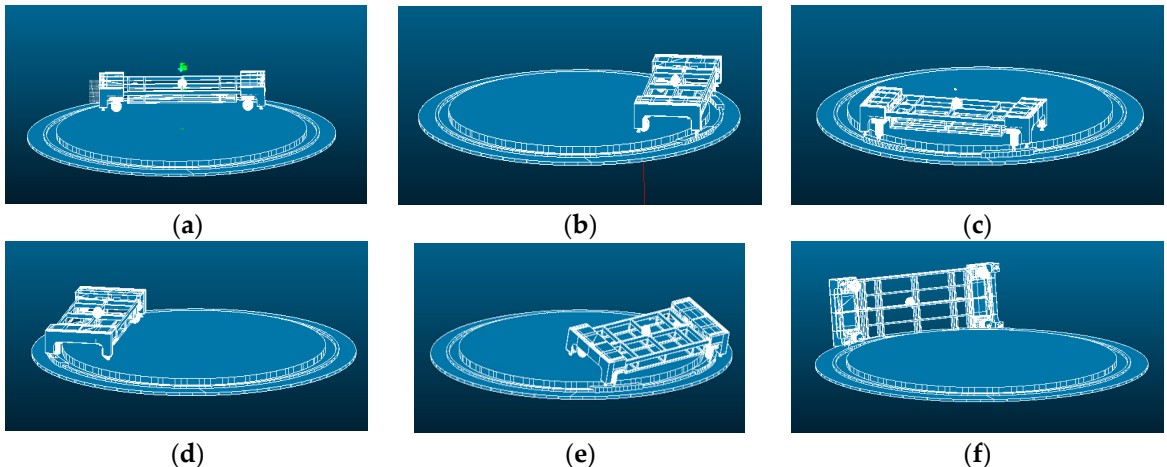

**Figure 6.** Simplified simulation model of heavy-duty AGV. (**a**) Simplified model position at 0 s. (**b**) Simplified model position at 2 s. (**c**) Simplified model position at 4 s. (**d**) Simplified model position at 6 s. (**e**) Simplified model position at 8 s. (**f**) Simplified model position at 10 s.

*3.4. Analysis of Factors Affecting the Rollover Stability of AGV*

3.4.1. Impact of Turning Speed on the Stability of Rollover in AGV

By comparing the movement of the AGV at different speeds, we can intuitively see the impact of the turning speed on the rollover of the AGV. The simulation control file is written in custom mode, setting the AGV's turning speed to start from 0 and accelerate uniformly to 100, with a linear velocity of 10.47 m/s. The running time and end conditions are defined on a 6 m radius bend with the AGV's centre of mass as the origin. The AGV is set to turn clockwise on the bend, with the left wheel as the outer wheel. The control file is imported, and the simulation is run, resulting in an AGV motion rollover lateral angle curve.

As shown in Figure 7, the effect of turning speed on the sideslip angle is shown. When the turning speed continues to increase to 87.3°/s, the value of the sideslip angle starts to increase significantly, and the AGV rollover occurs. The analysis proves that there is a regular relationship between the turning speed and the steering motion of the AGV.

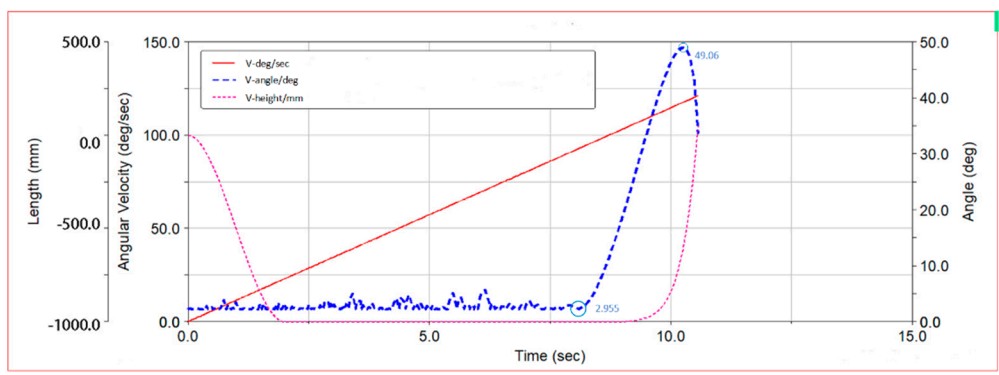

**Figure 7.** Graph of the Effect of Turning Speed Variation on the Sideslip Angle of the Centre of Mass.

3.4.2. The Impact of Centre of Gravity Position on the Stability of AGV Rollover

The centre of mass position of AGV is divided into three directions: normal, radial and tangential, and the three directions are the centroid height, the change of centre of mass

along the diameter direction and the motion tangent direction [19], and all three directions have different degrees of influence on the rollover of AGV.

As shown in Figure 8, the sideslip angle of AGV steering with different centroid heights is set to 1000 mm initially and increased by 200 mm each time. As the centroid height increases, the sideslip angle of AGV also increases linearly and regularly. For every 200 mm increase in centre of mass height, the lateral deflection angle increases by 1.12°. The waveform feature of the sideslip angle is more obvious when the centroid height increases gradually and reaches the maximum value at 4 s. The roll stability gradually decreases with the increase of centroid height.

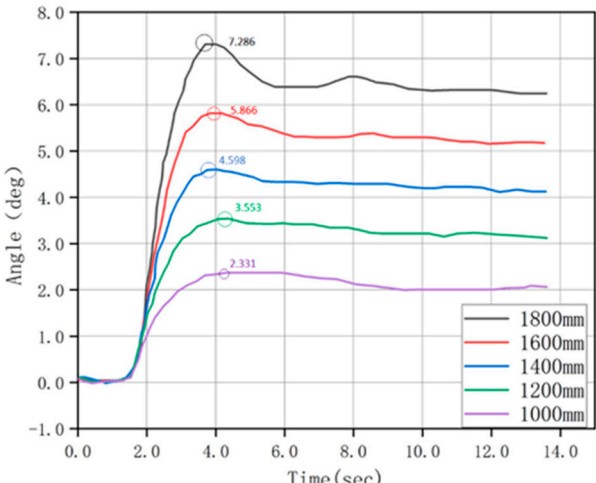

**Figure 8.** The lateral roll angle of AGV when the centroid height is changed.

Figure 9 shows the graph of the change of the sideslip angle of the centre of mass when the centre of mass position in the radial direction is changed. The sideslip angle increases significantly, but the increase is not significant. The sideslip angle increases to 2.04° for every 500 mm increase in the radial direction, which is linear with the sideslip angle.

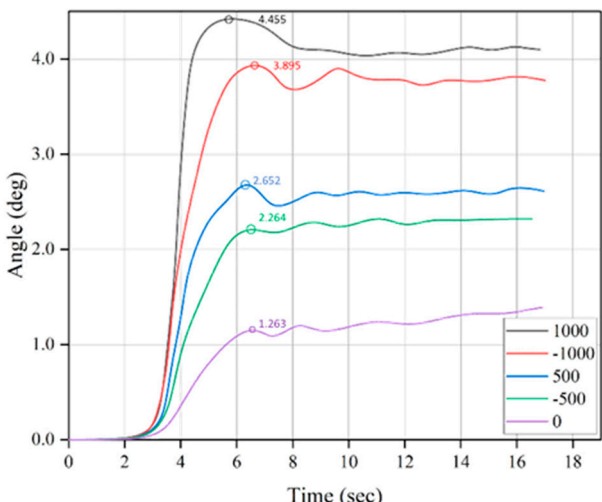

**Figure 9.** The lateral roll angle curves of AGVs at different radial direction positions.

As shown in Figure 10, the change of sideslip angle of the centre of mass when the tangential direction changes during the steering motion of AGV, the direction of motion is taken to be positive, and vice versa is negative, the centre of mass starts to move from −1000 mm positively at the zero moments in the tangential direction and reaches the minimum value when it approaches the centre of AGV at 8–10 s when the position of the

centre of mass continues to move positively along the tangential direction, the sideslip angle of the centre of mass continues to increase. It reaches the maximum value at 1000 mm. When the centre of mass position continues to move in the tangential direction, the sideslip angle increases, and the sideslip angle reaches the maximum at 1000 mm. In summary, the influence of the centroid height on the sideslip angle is the greatest in the three directions of the centre of mass position, followed by the radial direction and the smallest in the tangential direction.

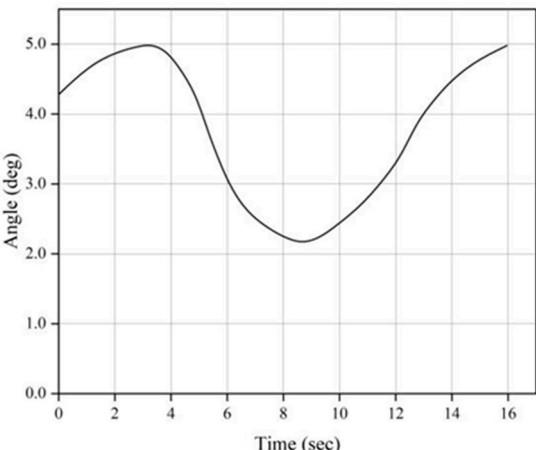

**Figure 10.** Curves of the effect of tangential direction change on the sideslip angle of the centre of mass.

### 3.4.3. The Influence of Road Friction Coefficient on the Stability of AGV Rollover

The road friction coefficient is a critical factor in the study of AGV rollover, and different road conditions have an important impact on steering safety. It takes the value of dry road friction coefficient as 0.8, road friction coefficient value as 0.6, rain road friction coefficient as 0.4, and an ordinary snow day as 0.2. The adhesion coefficient for compacted snow and near ice is 0.1 [20]. The outer wheel of the curved road of the AGV was used as the object of study to analyse the forces on the tires of the AGV under different road friction coefficient conditions. The road friction coefficient is 0.1, 0.2, 0.4, 0.6, 0.8; the turning speed is 70°/s; the turning radius is 6 m; the running time and end conditions of AGV are defined, and the control file is imported and simulated.

As shown in Figure 11, the lateral force curves of the outer wheels under different road friction coefficients are shown. The curves from bottom to top are the lateral force curves when the friction coefficients are 0.1, 0.2, 0.4, 0.6, and 0.8, respectively. When the AGV turns, the lateral force of the outer wheel rapidly increases along the reverse direction of the velocity as the speed increases. When the road friction coefficients are 0.6 and 0.8, the AGV will not slip, and the larger the road friction coefficient, the smaller the slipping trend. When the road friction coefficients are 0.1, 0.2, and 0.4, the maximum value of lateral force also increases with the increase of road friction coefficients, and the AGV is subjected to sliding friction. The maximum value increases with the increase of road friction coefficients, and the size of lateral force at this time is equal to the sliding friction.

The graph showing the change in the sideslip angle of AGV during a turning motion with different road friction coefficients is shown in Figure 12. The curves from top to bottom are road friction coefficients of 0.1, 0.2, 0.4, 0.6, and 0.8. It can be seen from the graph that as the road friction coefficient decreases, the sideslip angle of AGV during turning increases. At the same time, the fluctuations become more apparent, and the roll stability of AGV decreases.

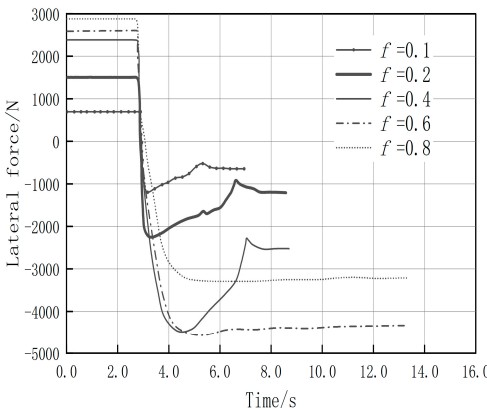

**Figure 11.** The lateral force curve of the outer wheel of AGV with the different road friction coefficients.

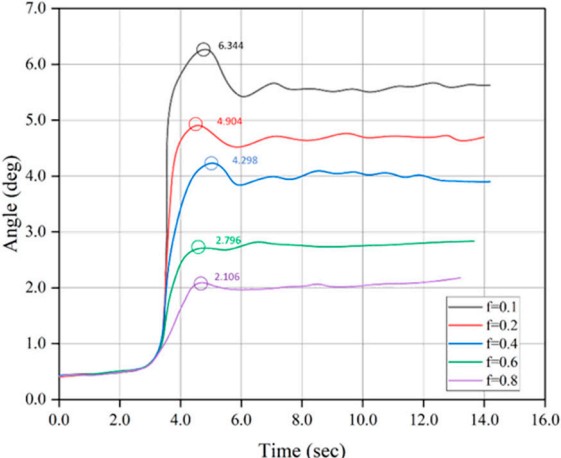

**Figure 12.** Variation of the side deflection angle of AGV with the different road friction coefficients.

## 4. Comprehensive Evaluation of Factors Affecting Rollover

### 4.1. Rollover Risk Metrics

In the evaluation of AGV rollover, the lateral-load transfer rate (LTR) index has higher credibility compared to the side acceleration and sideslip angle indices. Therefore, in order to study the rollover stability of heavy-duty AGVs, we usually use the lateral-load transfer rate (LTR) as an evaluation index for AGV rollover [21]. LTR can be represented by the following formula:

$$LTR = (F_1 - F_2)/(F_1 + F_2), \tag{32}$$

In the equation: $F_1$ is the vertical ground reaction force acting on the outer wheel of the AGV, and $F_2$ is the vertical ground reaction force acting on the inner wheel of the AGV.

The stability of heavy-duty AGVs during a rollover situation primarily depends on driving speed and road conditions. Our study has found that the load transfer rate (LTR) between the inner and outer wheels plays a crucial role in determining the rollover risk of an AGV. When the LTR equals 1, meaning that the vertical load on the inner wheel is zero, the AGV will experience a rollover. On the other hand, when the LTR is equal to 0, meaning that the loads on the inner and outer wheels are equal, the AGV will not experience a lateral transfer of load and, therefore, will not roll over. For values of 0 < LTR < 1, the AGV will undergo lateral load transfer, although it will not roll over, but as the lateral transfer becomes increasingly significant, the AGV will approach instability and will be prone to rollover [22].

As shown in Figure 13, to analyse the factors that affect the rollover of heavy AGVs during curve driving, a dynamic model of heavy AGVs during curve driving was established. Due to significant structural dimensions, high centre of gravity, and heavy load, the

established model is a rigid AGV model travelling on well-paved curved roads, ignoring the deformation of the AGV suspension and tires and rolling. The following results can be obtained by taking moments at point P on the AGV model.

$$\frac{1}{2}F_1B - \frac{1}{2}F_2B - ma_yh_g = 0 \tag{33}$$

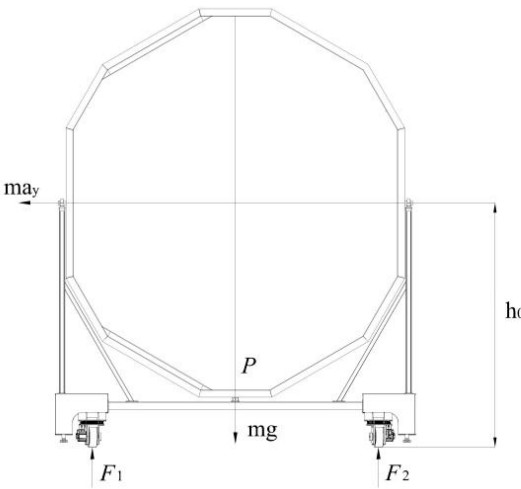

**Figure 13.** Steering dynamics model of heavy-duty AGV.

In the equation, $B$ is the wheel track, $a$ is the lateral acceleration of AGV, and $h$ is the height of the AGV centroid. The combination of (32) and (33) equations is:

$$LTR = \frac{2a_yh_g}{Bg} \tag{34}$$

When AGV is in curve motion, the lateral acceleration is related to the turning radius and driving speed of AGV:

$$a_y = \frac{v^2}{r} \tag{35}$$

In the equation, $r$ is the AGV steering radius, and $v$ is the AGV speed. Bring the above equation into the available:

$$LTR = \frac{2v^2h_g}{Bgr} \tag{36}$$

From Equation (36) can be seen: lateral-load transfer rate and AGV driving speed, the centre of mass high proportional; the greater the speed, the higher the centre of mass AGV curve driving the greater the possibility of a rollover. The transverse load transfer rate is inversely proportional to the wheelbase and steering radius; the larger the wheelbase and the larger the steering radius, the lower the possibility of overturning when the AGV is driving in a curve.

The curve of wheel lateral-load transfer rate (LTR) is output by ADAMS software, and the maximum value of LTR during steering is defined as LTRmax. The maximum value of LTR in the steering process of AGV is defined as LTRmax. LTRmax is used to describe the rollover tendency of AGV in the steering process, and LTRmax is used as a metric to measure the rollover risk of AGV.

### 4.2. Orthogonal Test Analysis of Rollover Influencing Factors

The method of the orthogonal experiment is adopted in this paper. Based on the principle of a single experiment, several main factors affecting the rollover of AGV are selected for orthogonal experiment analysis, which are the AGV turning speed (v), the

AGV centroid height (h), and the road friction coefficient (f). The target function of the entire rollover experiment research is chosen as LTRmax, and the orthogonal experiment analysis is carried out using the orthogonal table. Ignoring the mutual influence of each factor, the road friction coefficient for a dry road is 0.8, the road friction coefficient for a wet road is 0.6, the road friction coefficient for a rainy day is 0.4, and the road friction coefficient for a regular snow day is 0.2. According to the orthogonal experiment design scheme, the experiments are carried out one by one on the experimental platform, and the results are recorded. The experimental parameters are shown in Table 3.

**Table 3.** Lateral rollover stability factor level table.

| Level | Turning Speed V°/s | Centroid Height h/mm | Road Friction Coefficient f |
|:---:|:---:|:---:|:---:|
| 1 | 50 | 1200 | 0.2 |
| 2 | 60 | 1400 | 0.4 |
| 3 | 70 | 1600 | 0.6 |
| 4 | 80 | 1800 | 0.8 |

Through the analysis of 16 groups of experiments, the LTRmax value of each experiment and the average value $\bar{k}$ of 12 groups of index experiments were obtained, as shown in Figure 14. The Table S1 in Supplementary Materials.

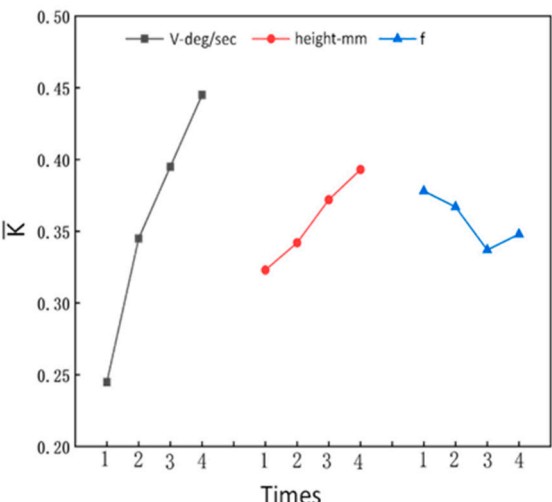

**Figure 14.** Range analysis diagram of the orthogonal test.

In Figure 14, the analysis of the orthogonal experimental results shows that among the three main factors affecting the stability of AGV rollover studied above, the effect of turning rate is the largest, followed by the height of the centre of mass. The most negligible effect on AGV rollover is the road adhesion coefficient. However, the road adhesion coefficient has the least effect, it is one of the main factors of AGV rollover, and the serious tendency of rollover can easily make the AGV trip and roll over. In order to validate the prediction accuracy of the mathematical model 36, the experimental parameters were inputted into the model, and the results obtained were compared with the experimental results. It was confirmed that the model could predict the rollover performance quite well.

### 4.3. Influence of Rollover Factors on the Load Transfer Rate

Taking the level of each factor as the abscissa and the average value of the test index (LTRmax) as the ordinate, the experiment was repeated five times, and the trend chart of factors and indicators was drawn. As shown in the following figure, the trend chart of factors and indicators can more intuitively see the trend of test indicators changing with factors.

The turning rate has a large influence on the lateral load transfer rate of the tires of AGV. The influence of the turning rate on the average value of LTRmax can be reflected in Figure 15. It can be seen that when the turning rate gradually increases, the average value of LTRmax increases and the influence becomes stronger and stronger, and the growth trend is almost linear. From Figure 16, it can be seen that the average value of LTRmax increases with the increasing height of the centre of mass, but when the height of the centre of mass exceeds 1600 mm, the slope of the curve becomes smaller. The rate of increase becomes smaller because the increase of the height of the centroid of mass makes the side turning force arm increase gradually, which increases the side turning moment of the AGV and the increase of the lateral load transfer rate of the AGV tires.

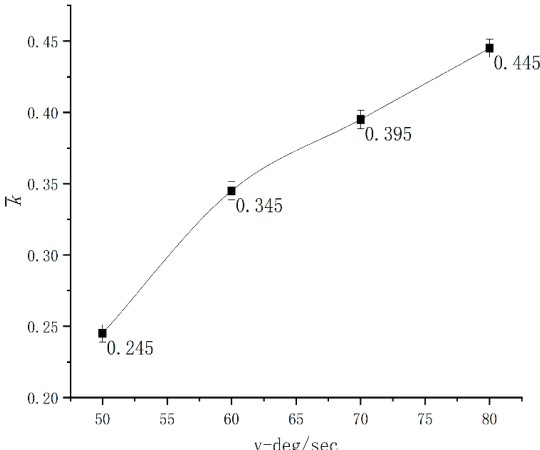

**Figure 15.** Effect of turning rate on the load transfer rate.

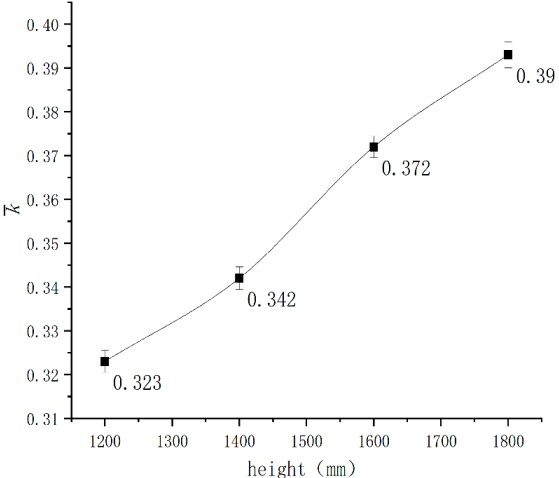

**Figure 16.** Effect of the centre of mass height on the load transfer rate.

As shown in Figure 17, as the road friction coefficient increases from 0.2 to 0.6, the average value of LTRmax decreases gradually. As the road friction coefficient increases to 0.6, the average value of LTRmax increases again. Within the range of road friction coefficient from 0.2 to 0.6, the lateral load transfer rate decreases from 0.378 to 0.337, meaning that the lower the road friction coefficient, the easier it is for the AGV to skid and if the skid is severe, it will result in a rollover. The AGV has a heavy load centre and a large rollover torque, decreasing rollover stability. Therefore, the average value is larger. As the road friction coefficient increases after reaching 0.6, the average value of LTRmax also increases. This is because when the road friction coefficient is higher, the AGV changes from skidding to rolling over, and if the speed is fast, the rollover stability decreases during the turn, and the AGV is more likely to roll over.

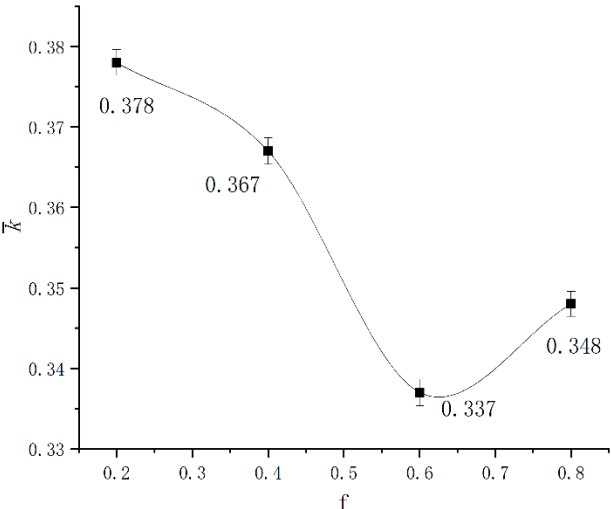

**Figure 17.** Effect of road surface adhesion coefficient on the load transfer rate.

### 5. Conclusions

In this study, the factors affecting the rollover stability performance of heavy-duty Automated Guided Vehicles (AGVs) were investigated through the use of ADAMS simulation software. The simulation results revealed that an increase in the turning rate leads to a decrease in rollover stability and an increased risk of rollover. The results also showed that when the turning rate is lower than 9.14 m/s, the rollover angle is relatively smooth, but if the safety value is exceeded, understeer occurs and leads to a rapid decrease in stability and an increased likelihood of rollover.

The influence of the center of mass position on roll stability was also analysed. It was found that the height of the center of mass has the greatest impact on roll stability, followed by changes in the radial direction and least by changes in the tangential direction. A 200 mm increase in the center of mass height was found to increase the lateral deflection angle of the AGV center of mass by 1.12°.

The road friction coefficient was also determined to be a significant parameter affecting side slip and roll over. When the road surface adhesion coefficient is below 0.6, the AGV is more prone to side slip and roll over. On the other hand, when the road surface adhesion coefficient is above 0.6, the mass side deflection angle increases with increasing road surface adhesion coefficient and turning rate, leading to an increased risk of rollover.

A proposed multivariate-based model for lateral load transfer rate is presented, which was used to study the effect of changes in turning rate, road adhesion coefficient, and center of mass height on rollover stability. By establishing this model, a method for assessing the rollover risk of platforms has been developed. This research has laid a solid foundation for studying the rollover stability of AGVs.

Finally, the study employed an orthogonal test method to analyse the relative impact of rollover stability factors and used the lateral load transfer rate as an evaluation index. The results showed that the turning rate had the greatest influence on rollover tendency, followed by the height of the center of mass, and the road adhesion coefficient had the least impact.

**Supplementary Materials:** The following supporting information can be downloaded at: https://www.mdpi.com/article/10.3390/lubricants11030119/s1, Table S1: Orthogonal test results.

**Author Contributions:** W.F.: conceptualization, methodology, investigation, writing. X.W.: validation, formal analysis, visualization, software. X.Z.: validation, formal analysis, visualization. investigation. All authors have read and agreed to the published version of the manuscript.

**Funding:** This research was funded by "BASIC AND APPLIED BASIC RESEARCH FUND OF GUANGDONG PROVINCE, grant number 2021A1515110927", "Jilin Province Scientific and Technological Development Program, grant number 20200201006JC", "The Open Project Program of Key Laboratory for Cross-Scale Micro and Nano Manufacturing, Minstry of Education, Changchun University of Science and Technology, grant number CMNM-KF202108", "Scientific Research Project of Education Department of Guangdong Province, grant number 2022KCXTD029".

**Data Availability Statement:** Not applicable.

**Conflicts of Interest:** The authors declare no conflict of interest.

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
