# Peer review of "Rollover Stability of Heavy-Duty AGVs in Turns Considering Variation in Friction Coefficient"

_lubricants, doi:10.3390/lubricants11030119_

Round 1
Reviewer 1 Report
This paper depicts the impact of turning rate, high center of mass, and adhesion coefficient in an Automated Guided Vehicle.
The paper is basically a case-study report where the variation of some parameters in the robotic platform is evaluated regarding the impact on the robot's performance.
Although the paper's purpose and development are clear, I advise making some additions and approaching changes that could help the paper to gain relevance in the research area:
1. The authors should propose a method or regulation that indicates the correct, efficient, or optimal use of the values of the different parameters according to their impact on the robotic platform's performance.
2. The paper should have a paragraph where the contributions of this research to state of the art in the area are named and explained.
3. A method or regulation should be proposed according to the contributions that the authors intend to make to the research area.
4. Since there is an in-depth study of the effects of parameter variability on robot performance, this study should be used to determine how effective the proposed system is.
5. Conclusions should focus on how the proposed policy, method, or use of parameters affects the robot's performance and how this proposed methodology presented better robot performance.
The general idea is to redirect the paper towards a more investigative approach.
Author Response
Q 1. The authors should propose a method or regulation that indicates the correct, efficient, or optimal use of the values of the different parameters according to their impact on the robotic platform's performance.
Answer 1: Thank you for your question. This article mentions in line 401 the establishment of a mathematical model for lateral load transfer, which determines the formula for the relationship between each variable and lateral load transfer rate. The study examines the degree to which each variable affects the lateral load transfer rate and obtains the optimal solution in the appendix.
Q 2. The paper should have a paragraph where the contributions of this research to state of the art in the area are named and explained.
Answer 2: Thank you for your correction on this issue. In conclusion, relevant models and contributions have been added in lines 491-493 of this article, making the entire conclusion more comprehensive and complete.
Q 3. A method or regulation should be proposed according to the contributions that the authors intend to make to the research area.
Answer 3: Thank you for raising this question. With regard to this issue, this paper describes in lines 380-383 that rollover occurs when the lateral load transfer rate is greater than 1, and based on this condition and the established mathematical model, a method for determining whether rollover occurs is provided.
Q 4. Since there is an in-depth study of the effects of parameter variability on robot performance, this study should be used to determine how effective the proposed system is
Answer 4: Thank you very much for raising this question. In response to this issue, this article has added a simple comparison in lines 435-436, making the overall structure more complete and providing guidance for future work.
Q 5. Conclusions should focus on how the proposed policy, method, or use of parameters affects the robot's performance and how this proposed methodology presented better robot performance.
Answer 5: Regarding this issue, this article has added a summary of a method for judging rollover at 490-491, which is based on the mathematical model used throughout the study to determine the level of influence of various variables on rollover.
Reviewer 2 Report
The manuscript meets the requirements of the journal and can be published with minor corrections:
The readability of figures should be improved.
In the introduction, the need for the conducted analyzes should be further substantiated and the introduction should be supplemented with a more detailed literature review.
Include references to all tables and figures in the text.
Explain the symbols in all formulas.
Author Response
Q1: The readability of figures should be improved.
Answer 1: All relevant contents in Figures have been modified.
Q2: In the introduction, the need for the conducted analyzes should be further substantiated and the introduction should be supplemented with a more detailed literature review.
Answer 2: The introduction section has been supplemented and revised, and the importance and significance of this study have been further elaborated. Please refer to the highlighted parts in the manuscript for details.
Q3: Include references to all tables and figures in the text.
Answer 3: The captions for the tables and charts referenced in the text have been added.
Q4: Explain the symbols in all formulas.
Answer 4: All symbols used in the equations throughout the article have been supplemented with their corresponding meanings.

Reviewer 3 Report
In the paper, the rollover stability analysis of the heavy duty AGV during turning was conducted. The Authors presented a kinematic and dynamic model of the AGV, then proposed a simplified simulation model in the computer physics environment. In the next step, Authors conducted and presented results of different simulations. They compared the influence of various parameters on the rollover stability using the lateral-load transfer rate (LTR) index.
This solution is very interesting, and therefore the paper is suitable for the journal. The scientific and engineering level of the paper is high. However, there are some major issues that prevent the article from being published in its current form.
1. There are parts of text i.e. subtitles with different fonts e.g: line 281, 282, 297, 331, 362, 428, 452.
2. Figures and charts have different styles, fonts, axis titles and colors. In some parts of the article, the descriptions are not readable:
a. Fig 1 – not readable, with some non-English words.
b. Fig 3 and Fig 4 are the same figure.
c. Figure 11 – axis titles not in English.
d. There are two Figures, with number 1, see page 16.
e. On page 16, there are two of the same figures.
f. Figure 15, 16 and 17 have different styling than previous part of the article.
3. Symbols and equation formatting are not consistent:
a. In line 75 there is symbol of “p”, but in Figure 1, the same symbol is marked as “P”.
b. Some symbols are once written as subscript but in another part of the article, the same symbol is written as a small letter, e.g: line 91 and Figure 1. Symbol hr, once written as subscript, once as a small letter.
c. In some parts of the article, symbols are written differently e.g: Table 1, velocity marked as capital V. In equation 24, the same velocity is marked as small v.
d. Equations 13 and 14 are edited differently, equation 14 seems to be stretched.
e. Equation 36 is stretched.
4. What does the sentence “First bullet” in line 290 mean?
5. What does the sentence “The standard English translation of this text is:” in line 363 mean?
6. Many misspelled words, e. g: in table 3 Tturning instead of Turning.
7. In table 3, some column names begin with an uppercase letter, some with a lowercase.
8. In line 443 the Annex 1 is mentioned, but not present in the article.
9. In line 442, Figure 15 and Figure 16, the K value is presented, but there is no explanation of what this symbol means and how it was calculated.
10. In line 497, Authors say that 200 mm increase in the center of mass height was found to increase the lateral deflection 497 angle of the AGV center of mass by 1.12°, but in line 313 the value is different.
11. The model used to present the kinematic and dynamic model is different to the model used in simulations.
12. Authors may consider adding input signals for simulation presented in Figure 7, 8, 9, 10. The verbal description is not clear.
The article can be accepted for publication after major revision.
Author Response
Q 1. There are parts of text i.e. subtitles with different fonts e.g: line 281, 282, 297, 331, 362, 428, 452.
Answer 1: Thank you for your reminder. The correction has been made in the text.
Q 2. Figures and charts have different styles, fonts, axis titles and colors. In some parts of the article, the descriptions are not readable:
- Fig 1 – not readable, with some non-English words.
- Fig 3 and Fig 4 are the same figure.
- Figure 11 – axis titles not in English.
- There are two Figures, with number 1, see page 16.
- On page 16, there are two of the same figures.
- Figure 15, 16 and 17 have different styling than previous part of the article.
Answer 2: Thank you for your reminder. The correction has been made in these Figures.
Q 3. Symbols and equation formatting are not consistent:
- In line 75 there is symbol of “p”, but in Figure 1, the same symbol is marked as “P”.
Answer 3a: Thank you for your reminder. The correction has been made in Figure 1.
- Some symbols are once written as subscript but in another part of the article, the same symbol is written as a small letter, e.g: line 91 and Figure 1. Symbol hr, once written as subscript, once as a small letter.
Answer 3b: Thank you for your reminder. The correction has been made in Figure 1.
- In some parts of the article, symbols are written differently e.g: Table 1, velocity marked as capital V. In equation 24, the same velocity is marked as v.
Answer 3c: Thank you for your reminder. The correction has been made in equation 24.
- Equations 13 and 14 are edited differently, equation 14 seems to be stretched.
Answer 3d: Thank you for your reminder. The correction has been made in equation 13 and 14.
- Equation 36 is stretched.
Answer 3e: Thank you for your reminder. The correction has been made in equation 36.
Q 4. What does the sentence “First bullet” in line 290 mean?
Answer 4: Thank you for your reminder. The correction has been made in equation 13 and 14.
Q 5. What does the sentence “The standard English translation of this text is:” in line 363 mean?
Answer 5: Thank you for your reminder. The correction has been made in text.
Q 6. Many misspelled words, e. g: in table 3 turning instead of Turning.
Answer 6: Thank you for your reminder. The correction has been made in text.
Q 7. In table 3, some column names begin with an uppercase letter, some with a lowercase.
Answer 7: Thank you for your reminder. The correction has been made in table 3.
Q 8. In line 443 the Annex 1 is mentioned, but not present in the article.
Answer 8: Thank you for bringing up this issue. The reference to the appendix will be added in the article.
Q 9. In line 442, Figure 15 and Figure 16, the K value is presented, but there is no explanation of what this symbol means and how it was calculated
Answer 9: Thank you for bringing this to our attention. The value of K is now included in the appendix and further explained in line 426.
Q 10. In line 497, Authors say that 200 mm increase in the center of mass height was found to increase the lateral deflection 497 angle of the AGV center of mass by 1.12°, but in line 313 the value is different.
Answer 10: Thank you for bringing this to our attention. We have made the following corrections in the text: added a sentence in line 310 stating that for every increase of 200mm in the height of the center of gravity, the roll angle increases by 1.12°, and in line 320 we clarified that the change in the lateral position of the center of gravity causes a change in the roll angle.
Q 11. The model used to present the kinematic and dynamic model is different to the model used in simulations.
Answer 11: Thank you for your correction. We did not simplify the kinematic and dynamic model analysis, but we simplified it in the simulation analysis. We treated the load on the AGV as a rigid body and set a center-of-mass sphere at the center-of-mass position of the object under study.
Q 12. Authors may consider adding input signals for simulation presented in Figure 7, 8, 9, 10. The verbal description is not clear.
Answer 12: Thank you for your correction. This article has been modified accordingly in the relevant section.
Reviewer 4 Report
The research analyzed the factors affecting the rollover stability of heavy-duty Automated Guided Vehicles (AGVs) through a multi-body dynamics simulation model. The study assessed the impact of turning rate, the center of mass height, and road adhesion coefficient on the AGV's rollover stability using a lateral deflection angle as a metric. The research found that the turning rate was the most critical factor affecting AGV rollover stability, followed by the center of mass height, while the road adhesion coefficient had the least impact. The study also derived a lateral-load transfer rate (LTR) index to evaluate the AGV's rollover behavior, considering the range of LTR values for different ranges of turning rate, the center of mass height, and road adhesion coefficient. The findings could help prevent AGV rollover incidents and enhance their safety. In general, the paper is well-written, but I have the following comments and questions that may be useful to improve the paper.
The simplifications made to the AGV model might not fully capture its real-world behavior and may introduce errors into the analysis.
The study only considers a number of factors (turning rate, centre of mass height, and road adhesion coefficient). What could be the effect of others that could affect the rollover stability of AGVs.
The study is based on simulation modeling, and the results may not necessarily generalize to real-world scenarios.
The use of a single metric (lateral deflection angle) to evaluate rollover behavior may not fully capture the complexity of the problem.
The intro can be (should be) extended to capture a better picture of the state of the art.
Author Response
Q1. The simplifications made to the AGV model might not fully capture its real-world behavior and may introduce errors into the analysis.
Answer 1: Thank you for your correction. It is true that there are often errors in most simulations. The simplification of the AGV model is aimed at obtaining results more efficiently and obtaining regularity results. In order to ensure the accuracy of the results, corresponding modifications have been made at line 270 and experimental analysis has been carried out.
Q2. The study only considers a number of factors (turning rate, centre of mass height, and road adhesion coefficient). What could be the effect of others that could affect the rollover stability of AGVs.
Answer 2:Thank you for providing your research ideas. The factors studied in this article are the most important ones that affect AGV, and other factors will also be analyzed in the future research process.
Q3. The study is based on simulation modeling, and the results may not necessarily generalize to real-world scenarios.
Answer 3:Thank you for your correction. The purpose of the simulation results in the article is to serve as a systematic study, and the quantitative results may have errors. In future research, we will try to minimize the errors.
Q4. The use of a single metric (lateral deflection angle) to evaluate rollover behavior may not fully capture the complexity of the problem.
Answer 4: Thank you for your feedback. This article evaluates the tipping behavior of AGV using two indicators: lateral tilt angle and lateral load transfer rate. This approach captures the complexity of the problem more effectively and is briefly introduced in line 373. By analyzing the tipping patterns of the variables using both indicators, we obtain good results. In future studies, we will introduce more evaluation indicators to make them more convincing.
Q 5. The intro can be (should be) extended to capture a better picture of the state of the art.
Answer 5:Thank you for your correction. The introduction from line 50 to 56 has been expanded to include more development status and corresponding references have been added.
Round 2
Reviewer 1 Report
Thank you for attending to the suggestions. However, I still believe that a paragraph should be included at the end of the introduction where the paper's contributions are listed and explained. (I understand that what I am asking is a formality, but it would help to understand the paper better)
Author Response
Dear Review
Thanks for your suggestion, I have added a contribution to the research in this paper in the last paragraph of the Introduction.
Weijie Fu
Reviewer 3 Report
The paper can be accepted.
Author Response
Dear Review
Thank you very much for your advice! Thank you for your recognition of the work in this article.
Weijie Fu